# One-Step Preparation of PVDF/GO Electrospun Nanofibrous Membrane for High-Efficient Adsorption of Cr(VI)

**DOI:** 10.3390/nano12183115

**Published:** 2022-09-08

**Authors:** Qingfeng Wang, Zungui Shao, Jiaxin Jiang, Yifang Liu, Xiang Wang, Wenwang Li, Gaofeng Zheng

**Affiliations:** 1Department of Instrumental and Electrical Engineering, Xiamen University, Xiamen 361102, China; 2Shenzhen Research Institute of Xiamen University, Shenzhen 518000, China; 3School of Mechanical and Automotive Engineering, Xiamen University of Technology, Xiamen 361024, China

**Keywords:** coaxial electrospinning, graphene oxide, Cr(VI) adsorption, heavy metal wastewater treatment

## Abstract

Mass loading of functional particles on the surface of nanofibers is the key to efficient heavy metal treatment. However, it is still difficult to prepare nanofibers with a large number of functional particle loads on the surface simply and efficiently, which hinders the further improvement of performance and increases the cost. Here, a new one-step strategy was developed to maximize the adhesion of graphene oxide (GO) particle to the surface of polyvinylidene fluoride (PVDF) nanofibers, which was combined with coaxial surface modification technology and blended electrospinning. The oxygen content on the as-prepared fiber surface increased from 0.44% to 9.32%, showing the maximized GO load. The increased adsorption sites and improved hydrophilicity greatly promoted the adsorption effect of Cr(VI). The adsorption capacity for Cr(VI) was 271 mg/g, and 99% removal rate could be achieved within 2 h for 20 mL Cr(VI) (100 mg/L), which was highly efficient. After five adsorption–desorption tests, the adsorption removal efficiency of the Cr(VI) maintained more than 80%, exhibiting excellent recycling performance. This simple method achieved maximum loading of functional particles on the fiber surface, realizing the efficient adsorption of heavy metal ions, which may promote the development of heavy-metal-polluted water treatment.

## 1. Introduction

Heavy metal pollution in water has become an environmental problem attracting worldwide attention because of its toxicity, harm to human health, and hidden danger to environmental safety. Among them, Cr(VI) is one of the most concerning pollutants produced by fuel combustion, tanning, electroplating, and the steel industry, with strong toxicity and carcinogenicity [1,2,3]. At present, chemical precipitation, membrane separation, ion exchange, electrochemical reduction, electrodialysis, photocatalysis, and adsorption have been used to remove Cr(VI) from an aqueous solution [4,5]. Compared with other technologies, the adsorption method is simple, is low−cost, uses fewer chemicals, is low-toxicity, and has been widely researched [6,7,8,9,10]. However, it still has disadvantages such as weak mechanical strength, possible secondary pollution, being difficult to recycle, and being easy to agglomerate [11,12]. Electrospun nanofibrous membranes (ENFM), as emerging adsorption materials, are widely used in the treatment of heavy metal pollution in water because of their highly specific surface area, simple preparation method, good permeability, and easy surface functionalization [13,14]. The satisfactory adsorption performance can be obtained through the optimization of membrane structure and the particle modification of the fiber surface [15].

Electrospinning blended with functional particles and polymer solutions is a common and simple method to realize the loading of functional particles. Wang et al. [16] utilized a electrpspun polyacrylonitrile (PAN) nanofibrous mat as template to obtain a polyacrylonitrile/polypyrrole core/shell structure for the Cr(VI) removal from aqueous solution. Wang et al. [17] used heat treatment to prepare a nanofiber membrane with an amine group-rich surface and an interconnection porous structure, which could achieve an adsorption capacity of 206 mg/g for Cr(VI) and remained effective after nine reuse cycles. However, blended electrospinning causes a large number of functional particles to be wrapped in the fiber, thus affecting the utilization of particles [18,19,20,21,22]. Coaxial electrospinning is an easy way to prepare core-shell structures. The sheath fluid can be changed into the dispersion of functional particles; thus, it is distributed on the surface of core fibers [23,24,25]. Our previous research also used the particle dispersion as the shell solution and prepared the functional electrospun fibrous membrane by coaxial electrospinning [26]. Due to the concentric spinneret, coaxial electrospinning allows functional materials contained in a sheath solution to be loaded on the nanofiber surface more directly and uniformly than electrospraying [27]. Xu et al. [28] fabricated porous polyacrylonitrile nanofibrous membranes through the combination of coaxial electrospinning and post-processing, achieving a uniform and large-scale distribution of nano-magnesium oxide on the fiber surface and good adsorption performance for Cu(II). However, there are few studies on the preparation of nanofibrous membranes for ion adsorption by combining coaxial electrospinning and blended electrospinning, which makes it possible to realize the fiber-surface loading of more functional particles in a simple one-step way.

Polyvinylidene fluoride (PVDF) is widely used as a substrate material because of its excellent electrospinning performance, stable mechanical properties, excellent resistance to high temperatures, and good chemical resistance [29,30]. Graphene oxide (GO) has excellent adsorption capacity for various ions due to its rich carboxyl, epoxy, hydroxyl, and other oxygen-containing functional groups [31,32]. Moreover, the addition of GO can also effectively improve the hydrophilicity of the ENFM, realizing the change of hydrophobic ENFM into hydrophilic one, thus improving the osmotic flux [33]. At present, GO has been well applied in the treatment of heavy metal pollution in water. Ren et al. [34] prepared PVDF–GO membranes by electrospinning with fixed nanoscale zero-valent iron particles, which were used to remove Cd-(II) and trichloroethylene pollutants in groundwater according to the gravity driven membrane filtration mechanism. Zhang et al. [35] prepared the PAN/GO solution using a mixture of N,N-dimethylformamide (DMF) and water as solvent and electrostatically spun it to obtain nanofibers with a Cr(VI) adsorption capacity of 382.5 ± 6.2 mg/g. Shraban et al. [36] prepared magnetic polyacrylonitrile-GO hybrid nanofibers by introducing magnetic Fe_3_O_4_ for Cr(VI) treatment in aqueous media, achieving a maximum adsorption capacity of 124.34 mg/g at pH = 3. Therefore, realizing a large amount of loading of GO on the surface of PVDF nanofibers will further improve the adsorption capacity of metal ions.

In this study, coaxial electrospinning and blended electrospinning were combined to prepare a large number of GO-loaded PVDF (PVDF/GO) nanofibrous membrane by one-step fabrication and realize the maximum GO loading that could not be achieved by coaxial electrospinning or hybrid electrospinning alone. The PVDF mixed with GO solution (PVDF–GO) was selected for the core layer solution, and GO dispersion solution was used for the shell layer solution, which could increase the GO load and effectively avoid the problem that the electrospinning process cannot be carried out due to the high GO concentration of the shell layer solution, achieving efficient metal ion adsorption. Moreover, the good hydrophilicity of GO particles and the excellent mechanical properties of PVDF nanofibers enabled the PVDF/GO ENFM to be quickly immersed and reused many times. Most importantly, this simple one-step strategy enables the low-cost and efficient preparation of nanofibers with a large number of functional particles on the surface, which further promotes the development and application of functional membrane materials.

## 2. Experimental

### 2.1. Materials and Chemicals

PVDF (*M*_W_ = 600,000) was purchased from Sigma-Aldrich (Shanghai, China). GO (the thickness was about 1 nm, the diameter of the lamella was 0.2–10 µm, the specific surface area was about 150 m^2^/g, and the oxidation level was 38%) was sourced from Tanfeng Tech. Inc. (Suzhou, China). DMF and potassium dichromate (K_2_Cr_2_O_7_) were purchased from Sinopharm Chemical Reagent Co., Ltd. (Shanghai, China).

### 2.2. Preparation of Electrospun Nanofibrous Membrane

PVDF/GO nanofibrous membranes were prepared by one-step coaxial electrospinning (i.e., PVDF@GO) combined with blended electrospinning (i.e., PVDF–GO).

#### 2.2.1. Preparation of PVDF ENFM

As shown in Figure 1a, PVDF powder was dissolved in DMF (10 g) solvent and stirred magnetic force for 12 h to prepare 12 wt% PVDF, denoted as PVDF. The solution was transferred to 1 mL syringe, and the flow rates was controlled by injection pump at 500 µL/h. The distance between the spinneret and the grounded collector was set to 15 cm, and a voltage of 20 kV was generated therebetween by a high-voltage power supply. The prepared nanofiber membrane was named PVDF.

#### 2.2.2. Preparation of Blended Electrospinning Nanofibers

As shown in Figure 1b, PVDF powder and GO powder of different qualities (0.1 and 0.2 g) were dissolved in DMF (10 g) solvent and stirred magnetic force for 12 h to prepare 12 wt% PVDF and GO (1, 2 wt%) mixed solution, denoted as PVDF–1GO and PVDF–2GO, respectively. The solution was transferred to 1 mL syringe, and electrospinning used the same parameters as PVDF. The prepared nanofiber membranes were named PVDF–1GO and PVDF–2GO, respectively.

#### 2.2.3. Preparation of Coaxial Electrospinning Nanofibers

As shown in Figure 1c, GO powder was added in DMF solvent to prepare 1 wt% GO dispersion and stirred at room temperature for 6 h to obtain uniform GO dispersion as sheath liquid. The PVDF solution was used as core liquid. The solution was transferred to two 1 mL syringes with the coaxial spinneret, and the flow rates of the internal and external liquid were controlled by two injection pumps at 500 µL/h and 8 µL/min, respectively. Other electrospinning parameters were consistent with the above experiment. The prepared nanofiber membrane was named PVDF@GO.

#### 2.2.4. Preparation of Coaxial Electrospinning Combined with Blended Electrospinning Nanofibers

As shown in Figure 1d, the uniform GO dispersion as sheath liquid and the above mixed solution (PVDF–1GO and PVDF–2GO) were used as core liquid. The solution was transferred to two 1 mL syringes with the coaxial spinneret, and electrospinning used the same parameters as PVDF@GO. The prepared nanofiber membranes were named PVDF–1GO@GO and PVDF–2GO@GO, respectively.

### 2.3. Morphological and Structural Characterizations

The surface morphology of different ENFM was analyzed by scanning electron microscopy (SEM, SUPRA 55 SAPPHIRE, Carl Zeiss AG, Jena, Germany) to observe and discuss the effect of GO loading on the nanofiber surface. The content of different characteristic elements in nanofibers was analyzed by energy dispersive X–ray spectroscopy (EDS, SUPRA 55 SAPPHIRE, Carl Zeiss AG, Jena, Germany). Fourier transform infrared (FTIR) spectrometer (NICOLET iS10, Thermo Fisher Scientifific, Waltham, MA, USA) was used to analyze and identify various functional groups.

### 2.4. Cr(VI) Adsorption Experiments

The K_2_Cr_2_O_7_ powder was dissolved in deionized water to prepare standard solutions of Cr(VI) with different concentrations. At 25 °C, 0.03 g dry PVDF–2GO@GO ENFM was directly immersed into 20 mL Cr(VI) solutions and shaken in a thermostatic shaker bath. The pH of the solution was adjusted by hydrogen chloride (HCl) and sodium hydroxide (NaOH). After a certain time, the supernatant was taken and the concentration of Cr(VI) in equilibrium solution was measured by UV spectrophotometer. Batch adsorption experiments were conducted to study the effects of solution pH, the initial concentration of Cr(VI), and the adsorption time on the adsorption capacity of ENFM. Adsorption capacity was determined by the following formula:(1)$$q_{e}=\frac{C_{0}-C_{e}}{M}V$$
where $q_{e}$ is the adsorption capacity (mg/g), $C_{e}$ and $C_{0}$ are the equilibrium concentration and initial concentration of Cr(VI) solution (mg/L), V is volume of the solution (L), and M is the mass of ENFM (g).

### 2.5. Adsorption Kinetics

By controlling the dosage of adsorbent, the initial concentration of solution, and the adsorption temperature, the relationship between the adsorption capacity of adsorbent and time was investigated. Pseudo-first-order and pseudo-second-order kinetic models were used to analyze its adsorption behavior. The linear forms of the two models were defined as
(2)$$\ln\left(q_{e}-q_{t}\right)=\ln q_{e}-k_{1}t$$
(3)$$\frac{t}{q_{t}}=\frac{1}{k_{2}q_{e}^{2}}+\frac{t}{q_{e}}$$
where $q_{e}$ and $q_{t}$ represent the amount of metal ion absorbed (mg/g) at equilibrium and at time t, respectively. $k_{1}$ and $k_{2}$ are the pseudo first-order and second-order rate constants, respectively.

### 2.6. Adsorption Isotherm

The results of Cr(VI) ion adsorption by PVDF–2GO@GO ENFM were analyzed using Langmuir isotherm and Freundlich isotherm, respectively. The expression for the Langmuir isotherm model was defined as
(4)$$q_{e}=q_{m}\frac{k_{L}C_{e}}{1+k_{L}C_{e}}$$
where $C_{e}$ represents the concentration of Cr(VI) ions (mg/L) at equilibrium, and $q_{e}$ and $q_{m}$ indicate the capacity at adsorption equilibrium and the maximum adsorption capacity, respectively. $k_{L}$ represent the Langmuir constant (L/mg).

The expression for the Freundlich isotherm model was defined as
(5)qe=kFCe1n
where $C_{e}$ is the equilibrium concentration of Cr(VI) solution (mg/L), n is the adsorption capacity, $q_{e}$ is the adsorption capacity at equilibrium (mg/g), and $k_{F}$ is the Freundlich adsorption coefficient.

### 2.7. Desorption Experiment and Reusability

In the desorption studies, NaOH solution was used as a desorption solution. The PVDF–2GO@GO ENFM was immersed into 30 mL NaOH (0.01 M) solution and shaken in a thermostatic shaker bath at 25 °C for 2 h. Before the next adsorption experiment, the nanofiber membrane was soaked in 0.1 M HCl solution. Then, the ENFM were washed repeatedly with deionized water and dried in an oven for 30 min. The experiments were repeated 5 times to verify the reusability of the prepared nanofiber membrane.

## 3. Results and Discussion

### 3.1. Design of Nanofibers to Maximize the Number of Surface Adsorption Sites

The adsorption of Cr(VI) is mainly achieved by the oxygen-containing functional groups on the surface of the nanofibers [31]. Many studies have adopted the method of blending to add adsorption functional particles, but only a small number of adsorption sites appear on the surface, and the utilization rate was insufficient. The coaxial electrospinning method enables more adsorption sites to be distributed on the surface of the nanofibers [24], effectively increasing the removal effect of harmful metal ions.

The processes of preparing nanofibrous membranes by different electrospinning strategies are shown in Figure 1. When only PVDF fibers were electrospun, there was no particle load on the fiber surface, and the fiber image was white, as shown in Figure 1a; when the mixed solution of PVDF and GO was electrospun, only a small amount of GO was supported because the excessive GO content would lead to high solution viscosity, and the membrane began to darken, as shown in Figure 1b [37]; as for the coaxial electrospinning only, the high content of GO in the shell solution would block the needle, while GO particles could be more loaded on the fiber surface without being embedded compared to hybrid electrospinning so that the fiber membrane became further black, as shown in Figure 1c [38]; they were all not conducive to the preparation of nanofibers with maximum GO load on the nanofiber surface. Thus, PVDF–GO nanofibers were prepared by coaxial electrospinning combined with hybrid electrospinning to maximize the adsorption sites on the nanofiber surface based on the experimental verification of the maximum acceptable amount of GO content in the shell and core layer solutions, which contributed to a highest GO content, and the membrane was the blackest, as shown in Figure 1d. Moreover, the hydrophilicity of GO also made it an effective strategy to effectively apply hydrophobic materials with stronger properties to water treatment.

### 3.2. Morphology and Chemical Characterizations of the Nanofibers

Figure 2 showed the FTIR spectra of GO, and the PVDF ENFM with different GO contents. The peaks at 1728 and 3350 cm^−1^ corresponded with the C=O and −OH stretching, respectively, which indicated the presence of carboxyl and hydroxyl functional groups in GO. The characteristic peak at 1624 cm^−1^ was due to the stretching of C=C in the phenol ring derived from the graphene oxide skeleton. For the PVDF membrane, the peaks at 1402 and 1172 cm^−1^ could arise out of the stretching and deformation vibrations of CH_2_ as well as the CF_2_ stretching vibration, respectively. These peaks also appeared in the spectrum obtained for the blended membrane. In addition, inside the dashed box is an enlarged view of the spectrum at 1649 cm^−1^. A new feature at 1649 cm^−1^ was seen in the spectrum of the PVDF and GO hybrid membrane when compared to that of the PVDF membrane, which was similar to the result of previous experiments [39,40]. The sharp absorption peaks at 602, 742, 838, and 1172 cm^−1^ can be attributed to the vibration absorption of the α-phase of PVDF, and the absorption peaks at 838 and 1280 cm^−1^ can be ascribed to the β-phase of PVDF. With the increase in GO load, all α-phase peaks decrease in intensity and β-phase peaks increased in intensity. The above results were similar to those of Chen et al. [20]. These results demonstrated that the large number of graphene oxide particles uniformly dispersed in PVDF enhances its α-phase to β-phase transition, due to the matching of the crystal lattice of GO with the β-phase of PVDF [20,41]. These results indicated the successful preparation of PVDF nanofibers loaded with GO.

Figure 3 shows the SEM images of the surface morphologies of nanofibers with different GO contents and different processes. It can be seen from Figure 3a that the electrospun PVDF nanofibers have a smooth surface. As shown in Figure 3b–f, after adding GO, the nanofibers exhibited a rough surface; the appearance of GO particles was observed on the surface of PVDF nanofibers; and the average fiber diameter became larger, which was due to the increase in solution viscosity caused by the addition of GO—thus the stretching of electrospinning jet was hindered [35,42,43]. The diameter of the PVDF, PVDF–1GO, PVDF@GO, PVDF–2GO, PVDF–1GO@GO, and PVDF–2GO@GO fibers were 337, 396, 490, 498, 444, and 649 nm, respectively. Moreover, as shown in Figure 1, it could be easily observed that the color of the prepared ENFM surface became darker as the increase in loading GO contents. Compared with the hybrid electrospinning only, the GO particles loaded on the surface of the nanofibers were significantly increased by coaxial electrospinning. This is also an effective way to solve the difficulty of GO wrapping during the electrospinning process [42].

As shown in Figure 4a, the content of oxygen element of different nanofibers was measured by EDS to quantify the adhesion of GO on the fiber surface. The picture showed the elemental mapping for the oxygen element of PVDF–2GO@GO ENFM. The oxygen element could be attributed to the existence of GO. With the increasing content of GO, the content of oxygen element increased from 0.44% to 9.32%; this also indicated that GO was successfully incorporated into PVDF nanofibers. In addition, when the same amount of GO was added, the content of oxygen element on the surface of the nanofibers prepared by coaxial electrospinning was also greatly improved compared with that of hybrid electrospinning, which reflected the advantages of coaxial electrospinning. For hybrid electrospinning, GO and PVDF were in a state of mutual mixing and were uniformly distributed in the nanofibers. After increasing the content of GO, there would be an increase in GO in the whole fiber (including the surface and inside of the fiber). After being combined with coaxial electrospinning, the GO dispersion of the shell layer was directly modified on the surface of the nanofiber of the core layer under the action of the coaxial spinneret, thereby increasing the GO content on the fiber surface. When the concentration of GO in the core layer solution was increased, the GO content on the surface of the nanofibers could be increased before the GO in the shell layer attaching, so that the GO content on the surface of the finally obtained nanofibers could be further improved.

The good wettability and immersing degree of ENFM have positive effects on efficient water treatment. Therefore, water contact angle tests were performed to explore the effect of GO content on the wettability and immersion degree of PVDF ENFM. Figure 4b illustrates the contact angles of different nanofiber membranes. The pristine PVDF ENFM exhibited higher hydrophobicity (140°) due to the intrinsic low surface energy of the fluorine in fluoropolymer [44], which had a weak affinity for water molecules; this prevented the penetration of water molecules into the ENFM [45,46], hindering immersion in liquid, as shown in Appendix A. With the increase in GO content, the water contact angle gradually decreased from 140° to 117° [39] and the state of immersion in solution became completely immersed. The full immersion of PVDF–2GO@GO ENFM in solution can make the adsorption site come into full contact with heavy metal ions, thus significantly improving the removal efficiency of Cr(VI). It was also an effective strategy to apply hydrophobic materials with excellent mechanical properties to water treatment, which could effectively improve the problem that the hydrophilic materials are easily damaged and difficult to treat using secondary treatment, due to long-term soaking in the water-treatment process.

### 3.3. Effects of Different Nanofibers and pH on Adsorption

Figure 5a showed the Cr(VI) removal effect of PVDF-based ENFM prepared by different GO contents and different methods within 2 h. The pristine PVDF membrane had a relatively low removal efficiency about 5.8%. With the increase in GO content, the removal rate in the same time also increased significantly. In addition, the PVDF@GO ENFM prepared by coaxial electrospinning also significantly improved the removal efficiency of Cr(VI) compared with the PVDF–1GO ENFM prepared by hybrid electrospinning. The PVDF–1GO@GO ENFM could reach the adsorption equilibrium faster than the PVDF–2GO nanofiber membrane prepared by hybrid electrospinning only. On this basis, the adsorption equilibrium of the PVDF–2GO@GO ENFM was reached within 1.5 h, and the removal rate reached 99%. According to the comparison of the adsorption effect of PVDF–1GO@GO ENFM and the previous four nanofibers, it could be seen that the combination of coaxial and hybrid electrospinning was an effective method and could reach the removal rate of more than 95% within 2 h. In addition, the adsorption time of PVDF–2GO@GO ENFM was only 1.5 h, indicating that we could accelerate the adsorption of Cr(VI) on the basis of this process method, to improve the adsorption effect in terms of efficiency. This also verified the effective increase in adsorption sites on the surface of nanofibers with the significant enhancement of the Cr(VI) removal ability.

The pH value of the solution also had a great influence on the adsorption and removal of Cr(VI) by ENFM [47]. The influence of solution pH value in the range of 1–7 on the removal of Cr(VI) during the adsorption time of 2 h by using the obtained PVDF–2GO@GO ENFM is shown in Figure 5b. With the increase in pH value, the adsorption capacity also decreased gradually. When the initial pH changed from 1 to 7, the removal percentage decreased from 99% to 1%. When pH values were lower than 6.8, the predominant species was hydrogen chromate (HCrO^4−^) coexisting with dichromate ions (Cr_2_O_7_^2−^). In an acidic media, more H^+^ ions accumulated on the adsorbent surface, and the protonated adsorbent in the PVDF–2GO@GO ENFM resulted in negatively charged Cr(VI) substances being attracted to the positively charged surface through electrostatic attraction. With more adsorption sites on the surface, the degree of protonation became higher, and the adsorption capacity was also significantly improved. The increase in pH led to a relative decrease in the protonation of the membrane surface, resulting in a weaker electrostatic attraction to the free oxygen anion and a decrease in the adsorption capacity [48]. When the pH > 6.8, the dominant species is chromate (CrO_4_^2−^). OH^−^ ions aggregated on the surface of the ENFM, resulting in its negative charge and electrostatic repulsion with negative Cr(VI) species, which greatly hindered the adsorption effect. Because PVDF–2GO@GO ENFM had the highest GO content, it was used in the next test. Generally, the pH values of electroplating and tanning wastewater are in the range of 2–5 and 7–8, respectively [49]. Among them, electroplating wastewater usually contains higher concentrations of Cr(VI) [50]. Therefore, subsequent experiments were carried out at the pH = 2.

### 3.4. Adsorption Efficiency and Kinetics

Figure 6 indicated the effect of 0.04 g PVDF–2GO@GO nanofibers on the removal of Cr(VI) from 35 mL of solution (100 mg/L) as a function of time. Obviously, during the initial time, the adsorption capacity increased more rapidly. After 240 min, the adsorption gradually reached the equilibrium point, and the adsorption capacity of PVDF–2GO@GO ENFM did not change significantly. The removal efficiency reached 99%, and the color of the solution changed from golden yellow to colorless.

In order to further explain the sorption behavior of Cr(VI) ions by PVDF–2GO@GO ENFM, the experimental adsorption data were analyzed using two commonly used kinetic models, namely, the pseudo-first-order and the pseudo-second-order [51]. The results of Cr(VI) adsorption on PVDF–2GO@GO ENFM fitted by the kinetic model are shown in Figure 7a,b and Table 1. The behavior of Cr(VI) adsorption is more in line with the pseudo-second-order model, as evidenced by the significantly lower R^2^-values of the pseudo-first-order model. This indicated that the adsorption process was restricted by chemisorption, which mainly involved sharing and transferring of electrons on the surface of PVDF–2GO@GO ENFM [52]. In addition, the pseudo-second-order kinetic model was also highly correlated, indicating the possibility that a certain physical adsorption process simultaneously exists to promote adsorption. The line graph of the pseudo-first-order kinetic model was plotted by $\ln\left(q_{e}-q_{t}\right)$ versus time, so the equilibrium adsorption amount $q_{e}$ must be obtained first. However, in the actual adsorption process, it took a long time to reach equilibrium, and the measured value of $q_{e}$ was not accurate enough. Therefore, it was often only suitable for the kinetic description of the initial stage of adsorption. After a period of adsorption, there will be a large deviation between the experimental data and the theoretical data [53].

### 3.5. Adsorption Isotherm

As shown in Figure 8, the adsorption capacity of PVDF–2GO@GO ENFM was also investigated at different initial concentrations of Cr(VI) in solutions ranging from 25 to 400 mg/L. However, at higher Cr(VI) concentrations, the saturation of the active sites resulted in a slow increase in adsorption capacity until the insignificant change of adsorption capacity.

The adsorption isotherm refers to the relationship between the residual ion concentration and the equilibrium adsorption capacity at a certain temperature. In this study, the Langmuir adsorption isotherm model and Freundlich adsorption isotherm model were used to further analyze the adsorption behavior of the PVDF–2GO@GO ENFM, where the Langmuir isotherm represented a single molecular layer adsorption, indicating that the individual adsorption sites of PVDF–2GO@GO ENFM were independent of each other and homogeneous. The Freundlich isotherm could be used to explain multilayer adsorption, where the heat of adsorption and affinity were unevenly distributed on a non-uniform surface [54].

Figure 9 shows the fitting curve of both the isotherm models, and Table 2 shows the model parameter values, which were calculated by the linear fitting equation. It could be seen that the correlation coefficient value of the Freundlich isotherm model equation for the adsorption of Cr(VI) by PVDF–2GO@GO ENFM was lower than that of the Langmuir isotherm model equation (0.9925 vs. 0.9808), which proved that the process was more suitable for analysis using the Langmuir model. Therefore, the adsorption behavior of the prepared nanofiber membranes for Cr(VI) relied mainly on the monolayer adsorption mode. There was no interaction between the adsorbates, and adsorption saturation was achieved when the adsorption sites were all occupied. The PVDF–2GO@GO ENFM had the characteristics of strong adsorption capacity, simple synthesis, and low cost, which further indicated its application potential in industrial wastewater treatment.

### 3.6. Regeneration Study of PVDF–2GO@GO Nanofibrous Membranes

Since the PVDF–2GO@GO ENFM had the best adsorption effect on Cr(VI) ions at lower pH values, the desorption of Cr(VI) could be achieved by alkali solution. In order to explore the recycling ability of PVDF–2GO@GO ENFM to adsorb Cr(VI) ions, five adsorption–desorption experiments were carried out successively. As shown in Figure 10a, the Cr(VI) adsorption capacity still remained above 80%.It can also be seen from Figure 10b that the nanofiber membrane after desorption was relatively well preserved, had good mechanical strength, and was easy to handle. The results showed that PVDF–2GO@GO ENFM had good reusability in the process of processing Cr(VI), as well as excellent recyclability. As shown in Table 3, the performance parameters of Cr(VI) adsorption in different types of literature were listed. Compared with the existing literature, the as-prepared PVDF–2GO@GO ENFM showed satisfactory results in terms of adsorption performance, but the step was simpler, which was beneficial to scale up production.

## 4. Conclusions

In this study, the PVDF–2GO@GO ENFM with maximum adsorption sites on the surface were obtained with a simple coaxial electrospinning and hybrid electrospinning synergistic process, which can effectively adsorb Cr(VI) ions in aqueous solution. The loaded GO nanoparticles obviously increased the oxygen content on the nanofiber surface from 0.44% to 9.32% and promoted the immersion process of the nanofiber membrane, which endowed the PVDF–2GO@GO ENFM with excellent adsorption capacity for Cr(VI) ions, reaching 271 mg/g. Moreover, the 99% removal rate for 20 mL Cr(VI) solution could be achieved within 2 h, and the used ENFM could effectively recover Cr(VI) for reusability. Thus, the coaxial electrospinning combined with the hybrid electrospinning process could effectively maximize the adsorption site content and utilization rate on the nanofiber surface, which greatly improved the water treatment effect. This strategy will further promote the application of functional particles on electrospun nanofibrous membranes and promote the development of water treatment.

## Figures and Tables

**Figure 1 nanomaterials-12-03115-f001:**
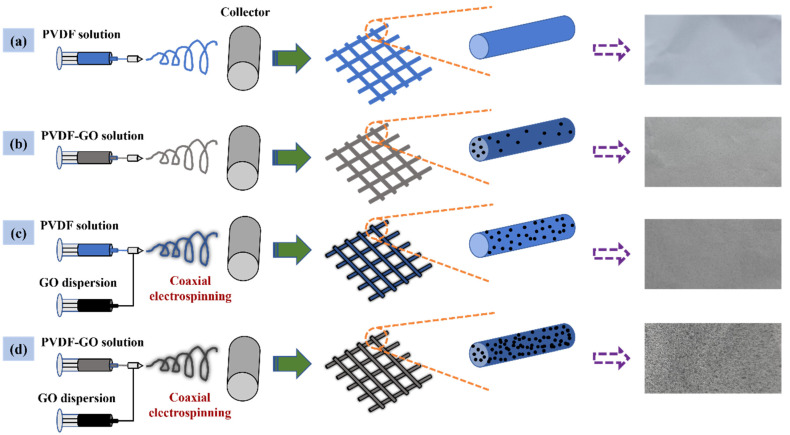
Schematic diagram of preparation by different electrospinning strategy: (**a**) preparation of original PVDF ENFM, (**b**) blended electrospinning, (**c**) coaxial electrospinning, and (**d**) coaxial electrospinning combined with blended electrospinning.

**Figure 2 nanomaterials-12-03115-f002:**
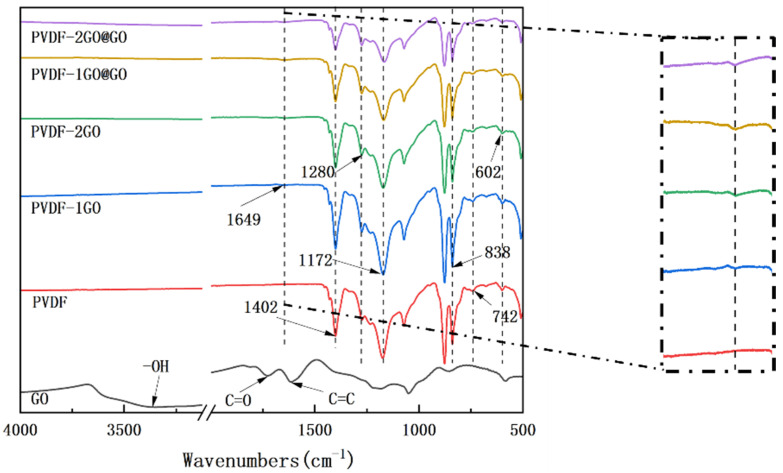
FTIR spectra of GO powder, PVDF, PVDF–1GO, PVDF–2GO, PVDF–1GO@GO, and PVDF–2GO@GO nanofibers.

**Figure 3 nanomaterials-12-03115-f003:**
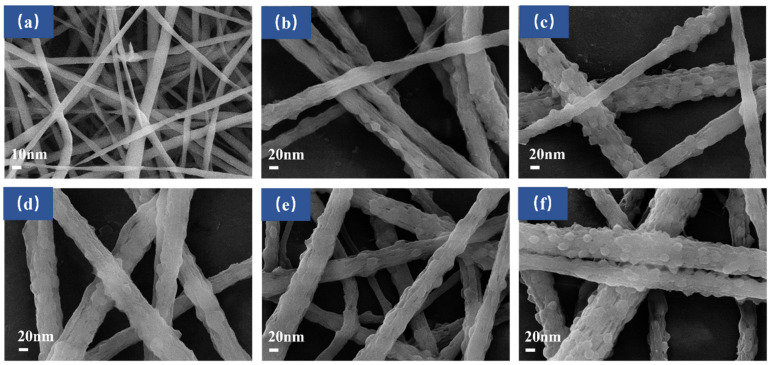
SEM of (**a**) PVDF (×10k), (**b**) PVDF–1GO (×20k), (**c**) PVDF@GO (×20k), (**d**) PVDF–2GO (×20k), (**e**) PVDF–1GO@GO (×20k), and (**f**) PVDF–2GO@GO (×20k).

**Figure 4 nanomaterials-12-03115-f004:**
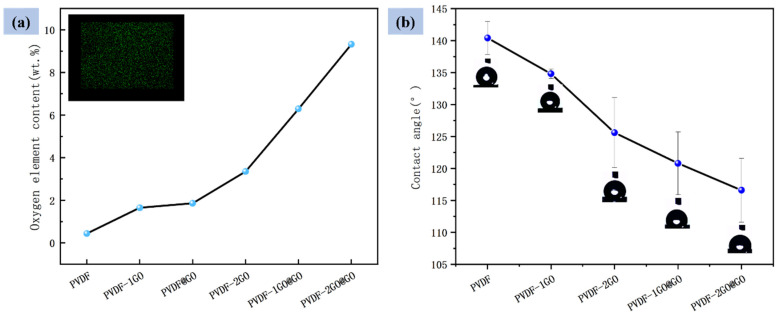
(**a**) Oxygen element content and (**b**) water contact angle of different nanofibers.

**Figure 5 nanomaterials-12-03115-f005:**
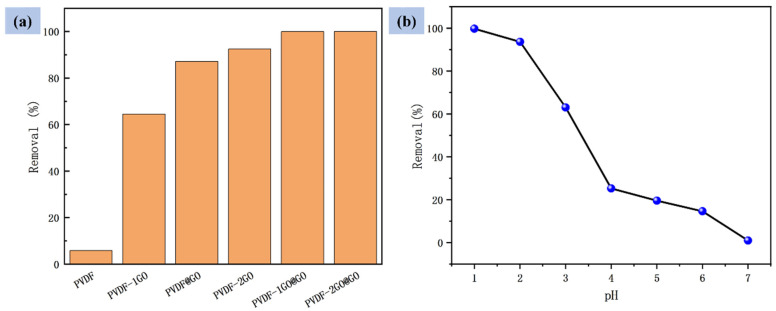
(**a**) Comparison of adsorption effect of different nanofibers and (**b**) effect of pH on the adsorption capacity by the PVDF–2GO@GO nanofibers.

**Figure 6 nanomaterials-12-03115-f006:**
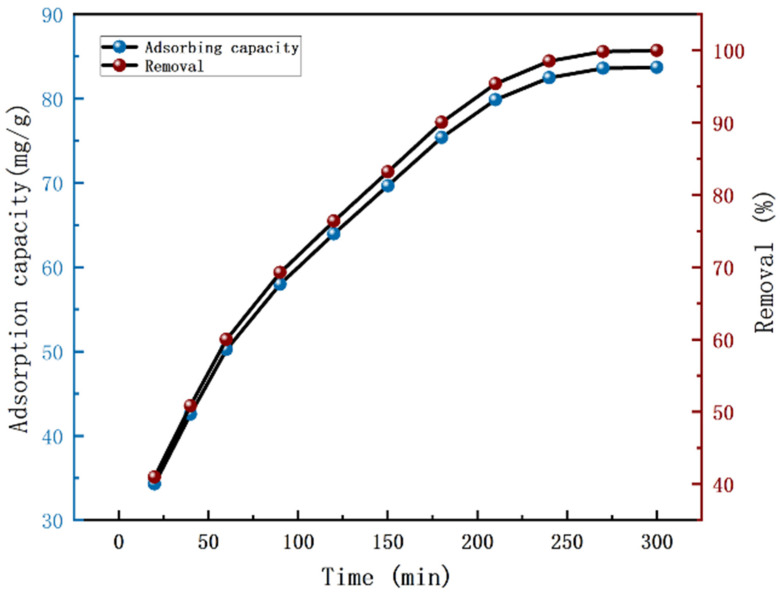
Adsorption capacity of PVDF–2GO@GO ENFM for Cr(VI) under different time.

**Figure 7 nanomaterials-12-03115-f007:**
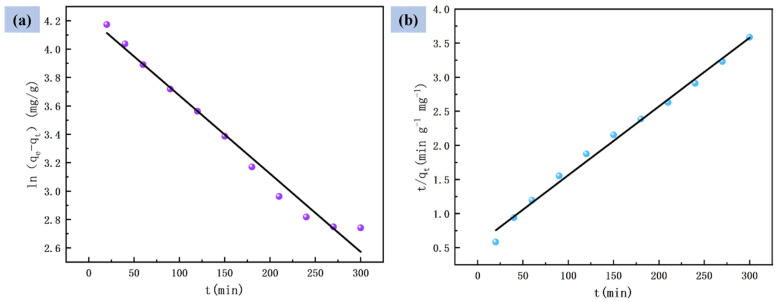
(**a**) Pseudo-first-order kinetic model and (**b**) pseudo-second-order kinetic model.

**Figure 8 nanomaterials-12-03115-f008:**
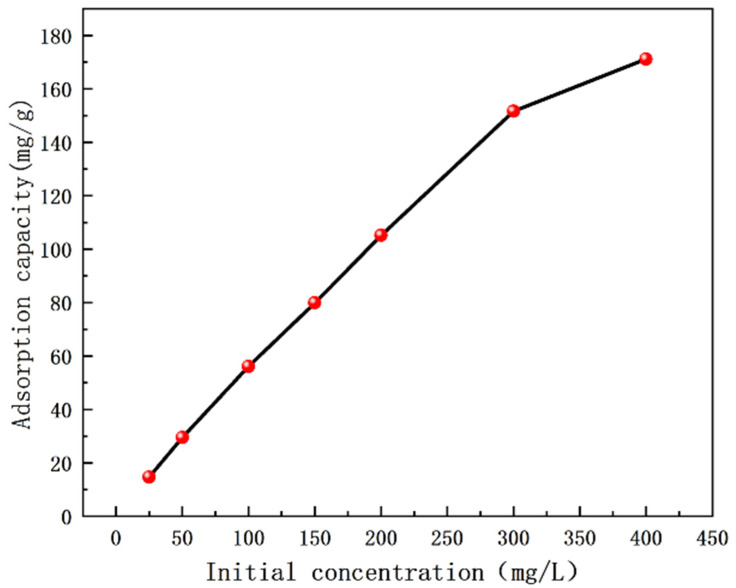
Effect of initial concentration of Cr(VI) aqueous solution on the adsorption capacity of PVDF–2GO@GO nanofibrous membranes.

**Figure 9 nanomaterials-12-03115-f009:**
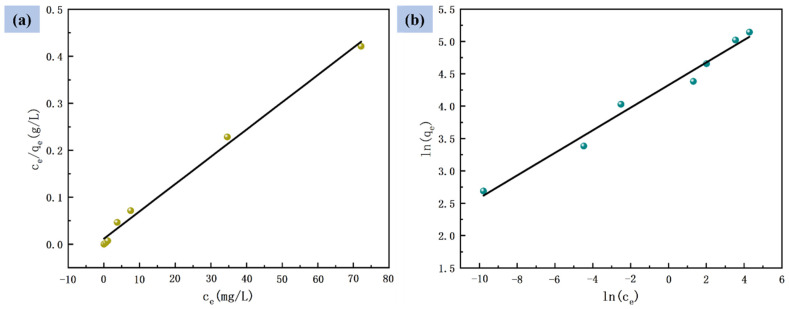
(**a**) Langmuir isotherm model and (**b**) Freundlich isotherm model for adsorption of Cr(VI).

**Figure 10 nanomaterials-12-03115-f010:**
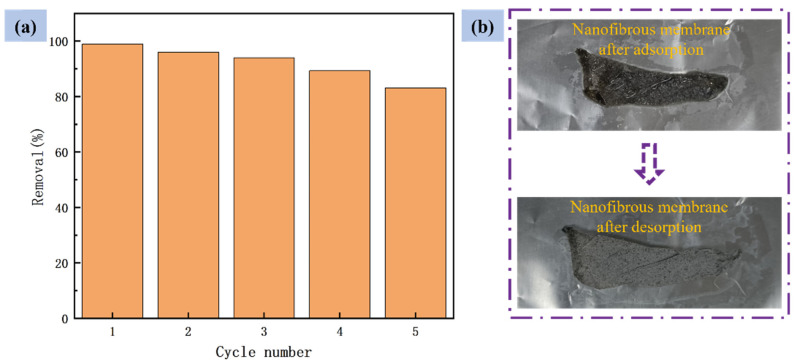
(**a**) Reusability of Cr(VI) from PVDF–2GO@GO nanofibrous membranes in cyclic adsorption–desorption test and (**b**) nanofibrous membranes after adsorption and desorption.

**Table 1 nanomaterials-12-03115-t001:** Comparison of kinetic parameters between the pseudo-first-order and pseudo-second-order models for Cr(VI) adsorption with PVDF–2GO@GO ENFM.

Model	Parameters	Values
Pseudo-first-order	*q_e_* (mg·g^−1^)	68.23
	*k*_1_ (min^−1^)	0.3830
	R^2^	0.9802
Pseudo-second-order	*q_e_* (mg·g^−1^)	99.21
	*k*_2_ (g·mg^−1^·min^−1^)	1.8314 × 10^−4^
	R^2^	0.9927

**Table 2 nanomaterials-12-03115-t002:** Comparison of kinetic parameters between the Langmuir and Freundlich models of Cr(VI) adsorption with PVDF–2GO@GO ENFM.

Model	Parameters	Values
Langmuir isotherm model	*q_m_* (mg·g^−1^)	172.12
	*k_L_* (L·g^−1^)	0.4878
	R^2^	0.9925
Freundlich isotherm model	*n*	5.7326
	*k_F_* (mg·g^−1^)	75.6509
	R^2^	0.9808

**Table 3 nanomaterials-12-03115-t003:** Comparison of adsorption performance of the PVDF–2GO@GO nanofibrous membranes with other adsorbents for Cr(VI).

Adsorbent	Adsorbent Quality (g)	pH	Equilibrium Time (h)	Maximum Adsorption Capacity (mg/g)	Ref.
GO–EDTA composite	0.05	1.8	12	37	[55]
Natural clay/Fe_3_O_4_/GO composite	1 g/L	3	1	71	[56]
GO-Fe_3_O_4_	0.1	2	1.4	3	[57]
NH_2_-GO decorated with Fe_3_O_4_ nanoparticles	0.2 g/L	2	12	123	[58]
PAN-GO-Fe_3_O_4_ composite nanofibers	0.06	3	1.1	124	[36]
PAN-NH_2_ nanofibers	0.025	2	1.5	137	[59]
Polyaniline-coated PVDF-HFP nanofibrous membranes	0.01	1.5	24	41	[44]
PAN/PPy core–shell structure nanofibers	0.1	2	12	75	[16]
PA 6/CS@Fe_x_O_y_ composite nanofibers	0.005	3	24	89	[60]
aminated-EVOH nanofiber membranes	0.05	2	8 d	235	[61]
Amidine PAN nanofibers	0.01	3	4	225	[62]
Chitosan/g-C_3_N_4_/TiO_2_ nanofibers	0.01	2	4	239	[63]
Porous PAN/GO nanofibers	0.05	3	1	382	[35]
PVDF–2GO@GO nanofibrous membranes	0.035	2	2	271	This study

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
