# Peer review of "One-Step Preparation of PVDF/GO Electrospun Nanofibrous Membrane for High-Efficient Adsorption of Cr(VI)"

_nanomaterials, 2022, doi:10.3390/nano12183115_

Round 1

Reviewer 1 Report

This manuscript reports the fabrication of PVDE/GO composite nanofibers with large loading of GO and the study of removal of Cr ions. The manuscript can be published with minor revision. 

The picture of the surfaces in Figure 1 can use the SEM images in Figure 3. The caption of Figure 4a should be Oxygen Element Content not O. Figure 7 needs more discussion. Figure 7a shows a deviation from linear relationship beyond 200 minutes. Is it because of the decrease of metal ion concentration?

The manuscript requires extensive editing. Some of the sentences were highlighted in the attached documents. 

Reviewer 2 Report

The present work reports the preparation of electrospun PVDF/GO materials and study of its adsorption properties for Cr(VI) ions. The article is a well-structured and is relevant to the Special Issue. However, there is a gap in knowledge identified, concerning coaxial electrospinning, and the terms for spinning solutions and spinning dispersions. The authors claim that have used coaxial surface modification technology and blended electrospinning. Unfortunately, the use of coaxial spinneret and combination of coaxial electrospinning with electrospinning, is not so simply and efficiently method for enrichment the fibers surface with GO. The point is that the experimental part must be carefully revised. For that reason, I consider that this paper needs some major amendments.

Specific comments and remarks: There are too many general mistakes concerning Experimental Part and lack of correct experimental information. For example, is the molecular weight of the PVDF correct 600 g/mol? There is no scientific explanation why the coaxial spinneret is used – the combination of electrospinning with electrospraying is more effective and simple method than using the combination of electrospinning with coaxial electrospinning. Moreover, the outer (sheath) solution is a GO dispersion in DMF which implies low viscosity and electrospraying instead of electrospinning. That is why, the experimental part, especially points 2.1., 2.2. and 2.3. must be rewrite. It will be more readable if the preparation of each membrane type shown in Figure 1 are explained separately in experimental part.

General questions

#1. GO is not soluble in DMF, so it is impossible to dissolve GO in DMF (line 105). Please use the correct explanation.

#2. The use of “@” symbol to denote mixtures or blends is not accepted. Please use the correct one.

#3. The sentence “The surface morphology of different ENFM was analyzed by scanning electron microscopy (SEM, SUPRA 55 SAPPHIRE, Carl Zeiss AG, Jena, Germany) to study the structure and distribution of GO on the surface of nanofibers” (line 117-119) is not correct. How exactly the distribution of GO onto fibers was studied by SEM?

#4. What is the mean diameter of the PVDF fibers and are they nano?

#5. Lines 189-190 – “oxygen–containing functional groups in GO” – which ones exactly?

#6. Figure 2 – for better clarity the FTIR curves should be given in the interval from 2000 – 500 cm-1

#7. Figure 3 – is the magnification of all SEM micrographs is equal? Please, add the magnification in the figure caption, as well what exactly the dashed rectangular is presented.

#8. Figure 4 (a) – the caption of Y axis is not correct – it is written “zero” instead of “O for oxygen”. What exactly the black picture is represented?

#9. Figure 4 (b) – what the picture is represented?

#10. Table 3 is redundant. It is more correct to compare only electrospun materials with GO.

#11. Conclusions must be shorter and more informative.

#12. References: The following article must be included and the results should be compared with:

Journal of Membrane Science 635 (2021) 119463, Graphene oxide functionalized polyvinylidene fluoride nanofibrous membranes for efficient particulate matter removal

Reviewer 3 Report

1. “The PVDF@2GO-GO ENFM had the characteristics of strong adsorption capacity, simple synthesis and low cost, further indicated its application potential in industrial wastewater treatment.” (Lines 340-341)

- What is usual pH value of the industrial wastewaters? What range of pH values could be used in industry during the treatment of wastewaters contaminated by Cr(VI)?

2. You have to state clearly and definitely the pH values applied during all of your adsorption experiments (Figs. 6-10 and Table 3). It seems that they were performed at very low pH values. If so, you have to explain in the text how these data could be relevant to the Cr(VI) removal from the industrial wastewaters.

 3. Table 3 (lines 362-363). Please add the information on the pH values used in the adsorption experiments in the cited papers. A correct comparison of the sorption capacity of nanofibrous membranes with other adsorbents is possible at the same or at least at the comparable pH values.

4. Experimental, line 96. More detailed characterization of the applied GO powder is needed (mean grain size, oxidation level etc.)

 5. Fig. 3. Are you sure that the order of 3c and 3d micrographs in this figure is correct?

 6. “In addition, the content of GO was increased as much as possible based on the combination of coaxial electrospinning and hybrid electrospinning, to maximize  the content of adsorption sites on the surface of nanofibers.” (Lines 219-221).

- Actually, it is not obvious from these micrographs. I would better skip this sentence.

7. Fig. 4a. How it may happen that the increase in the amount of GO in the core results in the increase in GO amount at the surface? How GO penetrates from the core to the surface through the sheath layer? Please explain it in the text.

 Why you didn’t increase the amount of GO in the sheath with little or no GO in the core composition?

 8. Fig. 4b. An immersion picture in the insert is too small and not so informative. I would better move it to the Supplementary.

9. “On this basis, the content of GO was successfully maximized by preparing the PVDF@2GO–GO ENFM, which showed the best adsorption and removal effect.” (Lines 262-263).

- It is difficult to agree with this statement, as the amount of GO at the surface of PVDF@1GO-GO and PVDF@2GO-GO is different indeed (Fig. 4a) while the Cr(VI) adsorption on these nanofibers is the same (Fig. 5a).

Round 2

Reviewer 2 Report

I accept in the present form